# SpeCache: Speculative Key-Value Caching for Efficient Generation of LLMs

**Shibo Jie** [1]   **Yehui Tang** [2]   **Kai Han** [2]   **Zhi-Hong Deng** [1]   **Jing Han** [3]

## Abstract

Transformer-based large language models (LLMs) have already achieved remarkable results on long-text tasks, but the limited GPU memory (VRAM) resources struggle to accommodate the linearly growing demand for key-value (KV) cache as the sequence length increases, which has become a bottleneck for the application of LLMs on long sequences. Existing KV cache compression methods, including eviction, merging, or quantization of the KV cache to reduce its size, result in irreversible information forgetting and potentially affect the accuracy of subsequent decoding. In this paper, we propose SPECACHE, which takes full advantage of the large and easily expandable CPU memory to offload the complete KV cache, and dynamically fetches KV pairs back in each decoding step based on their importance measured by low-bit KV cache copy in VRAM. To avoid inference latency caused by CPU-GPU communication, SPECACHE speculatively predicts the KV pairs that the next token might attend to, allowing us to prefetch them before the next decoding step which enables parallelization of prefetching and computation. Experiments on LongBench and Needle-in-a-Haystack benchmarks verify that SPECACHE effectively reduces VRAM usage while avoiding information forgetting for long sequences without re-training, even with a $10\times$ high KV cache compression ratio.

## 1. Introduction

The ability to handle long sequences is critical for large language models (LLMs), as it substantially impacts their

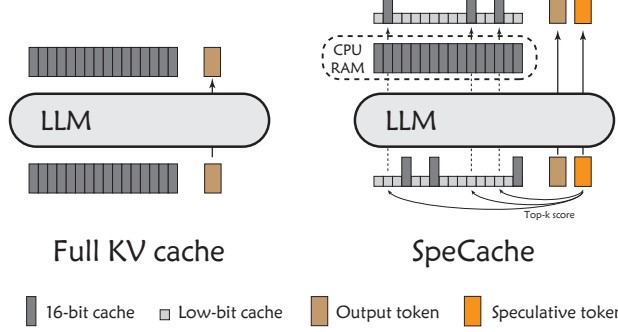

*Figure 1.* SPECACHE use low-bit KV cache and speculative token to "guess" the top-$k$ most relevant 16-bit KV pairs for the next token, and prefetch them before the next decoding step.

performance in tasks such as document processing, retrieval-augmented generation, and in-context learning. The commonly used transformer architecture in LLMs relies on key-value (KV) caches to avoid redundant computations during decoding. However, the size of the KV cache grows linearly with the sequence length, introducing significant memory overhead. For example, in the case of LLaMA 2-7B (Touvron et al., 2023) processing sequences of length 2k with a batch size of 16, the KV cache size reaches 8.4B, which exceeds the model's own parameter count. Since the memory of computing units (*e.g.*, GPU VRAM) is often limited, the KV cache becomes a bottleneck restricting the deployment of LLMs, especially on edge devices.

To alleviate the memory pressure caused by the KV cache, existing solutions, include using attention layers with slower KV cache growth (e.g., grouped query attention (Ainslie et al., 2023)) or attention mechanisms that do not require a KV cache (e.g., linear attention (Katharopoulos et al., 2020)). However, these methods alter the model architecture, necessitating re-pretraining of the LLMs. Another approach is post-training KV cache compression, including techniques like eviction (Zhang et al., 2023; Ge et al., 2024), merging (Zhang et al., 2024), and quantization (Liu et al., 2024b), but these methods are applied greedily during decoding and may result in the loss of crucial information for subsequent steps. Alternatively, KV cache can be offloaded to larger, more scalable memory (e.g., CPU memory or even disk), but this introduces frequent and substantial CPU-GPU communication, significantly increasing inference latency (Sheng et al., 2023).

---

[1]State Key Laboratory of General Artificial Intelligence, School of Intelligence Science and Technology, Peking University [2]Huawei Noah's Ark Lab [3]School of Artificial Intelligence, Beijing University of Posts and Telecommunications. Correspondence to: Zhi-Hong Deng <zhdeng@pku.edu.cn>, Jing Han <hanj@bupt.edu.cn>.

*Proceedings of the $42^{nd}$ International Conference on Machine Learning*, Vancouver, Canada. PMLR 267, 2025. Copyright 2025 by the author(s).

Existing research shows that the attention mechanism in LLMs is quite sparse (Liu et al., 2023b;a; Zhang et al., 2023), meaning that during decoding, we only need to ensure that a small number of KV pairs, which are most relevant to the current query, are fully present in VRAM. Based on this observation, the latency caused by offloading can be avoided via two strategies: *i)* Reduce the number of KV pairs prefetched to the GPU, fetching only several most important to the current query; *ii)* Perform the selection of important KV pairs far before the attention layer, enabling prefetching to run in parallel with GPU computation. However, achieving these two objectives without the need for retraining remains a challenge.

In this paper, we propose SPECACHE, a training-free method to implement the aforementioned strategies. SPECACHE stores the full KV cache in CPU memory. Before each attention computation begins, SPECACHE strives to ensure that the top-$k$ most relevant KV pairs has been already prefetched into VRAM. To achieve this, at each step, SPECACHE decodes two tokens simultaneously — an "output token" to compute the model's output, with the prefetched top-$k$ KV pairs, and a "speculative token" to guess the KV pairs most likely to be attended to in the next decoding step, with a low-bit copy of the KV cache in VRAM, as shown in Figure 1. These most relevant 16-bit KV pairs for the next step are prefetched in parallel into VRAM before the attention computation of the following step.

Since the decoding process of LLMs is memory-IO bound, GPU utilization is quiet low, meaning that decoding two tokens in parallel introduces almost no additional latency. Moreover, because we can prefetch the necessary KV pairs for attention one step ahead, prefetching and computation can occur in parallel, avoiding any increase in inference latency. We conducted experiments on various LLMs using the LongBench (Bai et al., 2024) and Needle-in-a-Haystack benchmarks (Greg Kamradt, 2023). The results demonstrate that, compared to existing KV cache compression methods, SPECACHE achieves better performance with a even smaller KV cache size. Moreover, the performance of SPECACHE is nearly on par with that of the original KV cache, even with only 10% KV cache size. Benefiting from this, SPECACHE can increase the batch size of decoding by up to $12\times$, achieving $4.6\times$ larger throughput compared to the original KV cache.

## 2. Related Work

### 2.1. Efficient KV Cache

Existing work on optimizing the KV cache size during the inference process of transformer-based LLMs can be categorized into the following three directions:

**KV-Efficient Architecture.** The size of the KV cache is directly determined by the model architecture. By modifying the model structure, the size of the KV cache can be reduced. Multi-Query Attention (MQA) (Shazeer, 2019) shares the key and value across all attention heads, while Grouped-Query Attention (GQA) (Ainslie et al., 2023) groups attention heads and shares the key and value only within each group. YOCO (Sun et al., 2024b) allows the latter half of the layers in LLMs to reuse the key and value computed by the earlier half of the layers. Multi-Head Latent Attention (MLA) (DeepSeek-AI et al., 2024) re-parameterizes the key and value as linear projections of low-rank space vectors. Additionally, some approaches modify the attention mechanism to avoid the linearly growing KV cache, such as RWKV (Peng et al., 2023), RetNet (Sun et al., 2023), and State Space Models (Gu & Dao, 2023). *These methods alter the model architecture and therefore must be applied before pre-training begins, making them unsuitable for training-free optimization during the inference stage of off-the-shelf LLMs.*

**Post-Training Compression.** Existing post-training KV cache compression methods include eviction, merging, and quantization of KV pairs. StreamLLM only retains the most recent KV pairs and a few initial KV pairs. H2O (Zhang et al., 2023), Scissorhands (Liu et al., 2023a), and RoCo (Ren & Zhu, 2024) use attention scores to measure the importance of KV pairs and greedily drops unimportant pairs. FastGen (Ge et al., 2024) introduces four policies to determine which KV pairs to keep and selects the optimal policy combination for each attention head during the prefilling stage. CaM (Zhang et al., 2024) and D2O (Wan et al., 2024) selectively merge the KV pairs that are about to be evicted with those that are being retained, thereby preserving partial information. MiniCache (Liu et al., 2024a) leverages the similarity of KV caches between adjacent layers to merge them across layers. KIVI (Liu et al., 2024b) and KVQuant (Hooper et al., 2024) apply per-channel key quantization and per-token value quantization to compress the KV cache down to 2-bit. ZipCache (He et al., 2024) introduces channel-separable token-wise quantization to achieve an even higher compression rate. These methods can reduce the KV cache size in a training-free manner during the inference stage. *However, since compression inherently leads to information loss, greedy compression at the current step may discard information that could be useful for future steps, potentially degrading the performance of LLMs.*

**Offloading & Prefetching.** FlexGen (Sheng et al., 2023) offloads the KV cache to CPU memory or even disk and searches for the optimal offloading strategy. Huggingface's `transformers` (Wolf et al., 2019) library also implements a simple offloaded KV cache. InfLLM (Xiao et al.) segments long-range context into memory units that are offloaded, and during inference, it retrieves and loads only the

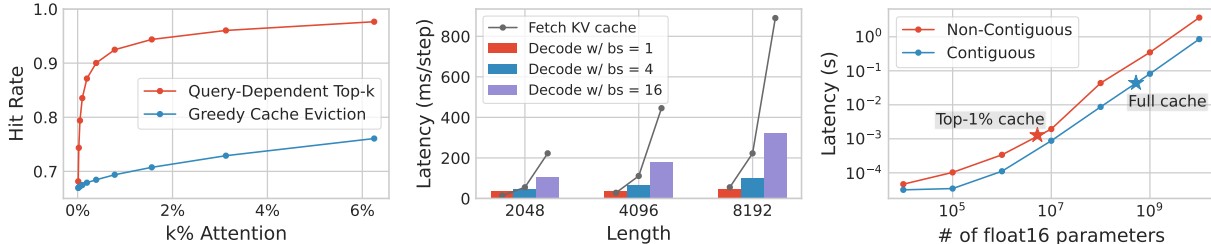

*Figure 2.* ***Left:*** Hit rates of query-dependent top-$k$ attention and greedy cache eviction. ***Middle:*** The latency of a single decoding step on the GPU *vs* the latency of loading the KV cache from the CPU to the GPU. ***Right:*** CPU-GPU transfer latency of contiguous and non-contiguous CPU memory. We highlight the full and top-1% KV cache size with context length of 32k. Measured using Mistral-7B-Instruct-v0.2 on NVIDIA A6000 GPU.

units relevant to the current token for attention computation. *While these methods reduce VRAM usage without losing KV cache information, the frequent CPU-GPU communication significantly increases inference latency.* In response to this, ShadowKV (Sun et al., 2024a) also tries to reduce the amount of fetched parameters to reduce latency.

### 2.2. Bottleneck of LLM Inference

The inference process of LLMs can be divided into two phases: the prefilling phase, where the model generates the KV cache for the prompt and produces the first output token, and the decoding phase, where the output token from the previous step serves as input to generate the next token. Generally, the bottleneck in the prefilling phase is the computational speed of the GPU, making it computation-bound, while the bottleneck in the decoding phase is the speed of data transfer between High Bandwidth Memory and Static Random Access Memory, making it memory-IO bound. Many techniques, such as batching (Kwon et al., 2023) and speculative decoding (Leviathan et al., 2023), leverage this characteristic by simultaneously inputting multiple tokens during decoding, which, although increases FLOPs, enhances GPU utilization and mitigates significant increases in single-step latency, and ultimately enhances overall throughput.

## 3. Method

### 3.1. Profiling the Sparsity of KV Cache

Existing work has already explored the sparsity of attention in LLMs (Tang et al., 2024; Singhania et al., 2024). We begin by investigating the extent of this sparsity and the potential efficiency gains from transferring only the sparse KV cache.

We conducted experiments on the LLaMA-3-8B model using the PG19 dataset truncated to a sequence length of 8196. We measured the hit rate, which represents the proportion of attention scores captured by two types of sparse attention

compared to full attention scores. The two sparse attention mechanisms are: *i)* Query-dependent top-$k$ attention, where each query only includes the KV pairs with top-$k$ attention scores in its attention computation. *ii)* Greedy eviction, similar to H2O, where KV pairs with low cumulative attention scores are evicted to ensure that each query attends to only $k$ KV pairs.

Based on the results shown in Figure 2 (left), we can conclude that: *i)* Attention is highly sparse. Only 0.5% of the keys can cover 90% of a query's attention. *ii)* The sparsity of attention is query-dependent. While both methods enforce the same level of sparsity, the hit rate for greedy eviction is much lower than that of query-dependent top-$k$ attention. This suggests that different queries tend to focus on distinct sets of keys. Although greedy eviction optimizes for the current query, it fails to preserve important KV pairs for subsequent queries, potentially evicting KV pairs that are crucial for future steps. Therefore, while eviction methods can achieve sparse attention, they cannot recover evicted KV pairs, resulting in lower hit rates. Consequently, dynamic prefetching is crucial for maintaining attention performance.

### 3.2. Profiling the Efficiency of KV Cache Offloading

As shown in Figure 2 (middle), simply offloading and prefetching the entire KV cache leads to substantial CPU-GPU transfer overhead. The experiments above inspire us to prefetch only a small number of the most important KV pairs during each decoding step, thereby reducing the number of transferred data. For non-contiguous memory transferring, mainstream framework such as PyTorch introduces additional time overhead. Therefore, we need to conduct experiments to verify the efficiency advantage of transferring sparse KV cache compared to the full KV cache.

As shown in Figure 2 (right), although sparse CPU-GPU transfers can introduce approximately 5 times the latency, we can enhance efficiency by reducing the amount of data transferred, since the attention mechanism maintains a high hit rate even at 1% sparsity. For instance, with Mistral-7B-

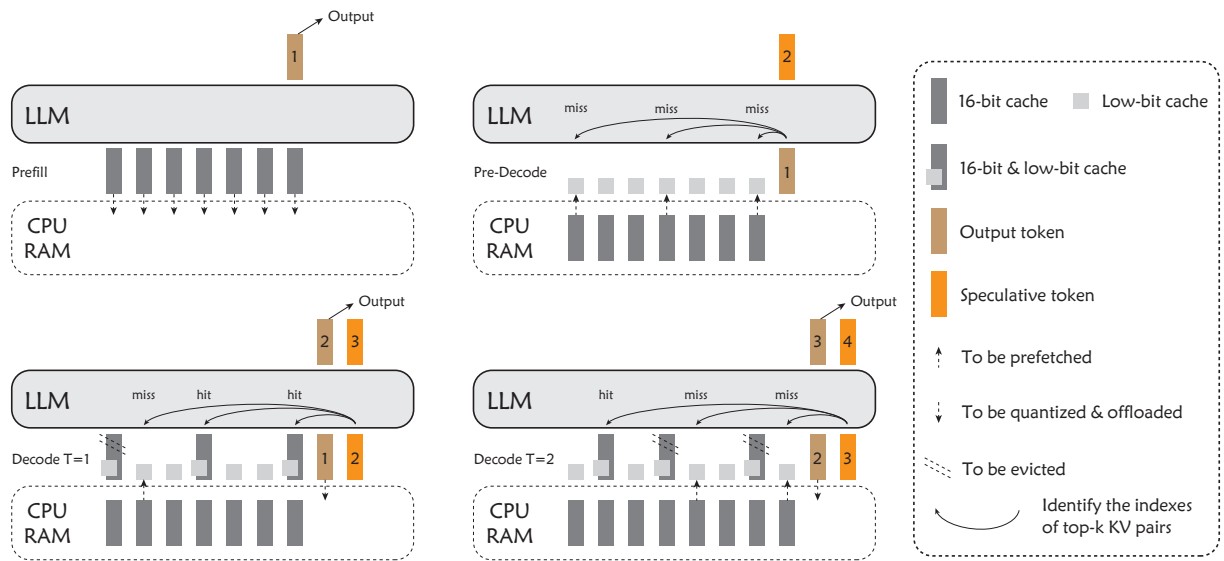

*Figure 3.* **Illustration of SPECACHE.** In prefilling stage, the KV cache is quantized and offloaded layer by layer. In pre-decoding stage, we use the first output token to compute the first speculative token and prefetch the 16-bit KV pairs needed by the first decoding step. In each step of decoding stage, we simultaneously decode two tokens: the output token and the speculative token. the results of both serve as inputs for the next step. The top-$k$ most relevant 16-bit KV pairs for the speculative token are prefetched before the next step.

Instruct-v0.2 model and context length of 32k, transferring only top-1% of the KV pairs would reduce transfer latency by 95%.

### 3.3. SPECACHE: Speculative Key-Value Caching

Although the efficiency benefits of top-$k$ prefetching are significant, we still need a method to predict the attention scores of KV pairs before loading the KV cache. To enable parallel prefetching and computation, we need to begin prefetching the KV pairs as early as possible. This brings us to a key challenge: **How can we determine which KV pairs are important far before the attention operation takes place?** In fact, we don't need to prefetch the exact top-$k$ KV pairs; we only need to prefetch KV pairs with a high hit rate, ensuring that they include the vast majority of significantly attended ones. Based on this, we propose SPECACHE, a method that speculatively predicts which KV pairs will be important for future queries and prefetches them accordingly.

Speculatively predicting future attention scores requires approximate representations of both the historical key cache and the future query to be available in VRAM. For the former, existing training-free KV cache quantization methods can effectively address this issue; for example, we can store a 2-bit or even 1-bit approximation of the KV cache in VRAM. As for the latter, we propose decoding an additional "speculative token" in parallel in each decoding step to approximate the next token.

As shown in Figure 3, The entire inference process can be divided into three stages: prefilling, pre-decodeing, and decoding.

**Prefilling.** During the prefilling stage, we adopt a layer-by-layer offloading approach. After the computation of each attention layer is completed, we quantize the KV cache to low precision and offload the original 16-bit KV cache $C$ entirely to CPU memory, freeing up space for the next layer's KV cache. Once the prefilling stage is complete, we obtain an accurate first output token $T_1$.

**Pre-decoding.** Before the decoding stage begins, only the low-bit KV cache $C'$ resides in VRAM. To prefetch the necessary 16-bit KV pairs for the first decoding step, we introduce a single decoding step as pre-decodeing. We use $T_1$ as input, generating a preliminary output $T'_2$, which may not be fully accurate, as a speculative token. Simultaneously, we record the indexes of top-$k$ KV pairs from the attention score calculation for $T_1$, denoted as $\mathcal{K}_1$, and immediately start prefetching those 16-bit KV pairs from CPU RAM in parallel with the computation of subsequent layers. After the pre-decoding step concludes, the first decoding step follows immediately.

**Decoding.** Before the $t$-th decoding step begins, VRAM contains the previous output token $T_t$, a speculative token $T'_{t+1}$, low-bit KV cache $C'$, and the top-$k$ 16-bit KV pairs $C_{\mathcal{K}_t}$ required by $T_t$.

We decode $T_t$ and $T'_{t+1}$ in parallel,

$$\boldsymbol{O} = \texttt{Attn}([T_t, T'_{t+1}], \boldsymbol{C}' \cup \boldsymbol{C}_{\mathcal{K}_t}) \tag{1}$$

in which $\boldsymbol{C}' \cup \boldsymbol{C}_{\mathcal{K}_t}$ denotes replace $\boldsymbol{C}'_{\mathcal{K}_t}$ (the subset of $\boldsymbol{C}'$

*Table 1.* **Performance of SPECACHE across various LLMs on LongBench.** "16-bit" denotes vanilla KV cache without reduction technique, "2-bit" denotes 2-bit KIVI, and "1-bit" denotes KIVI modified for 1-bit quantization. The KV cache size is calculated based on the models' maximum context length.

| | Bit-width of KV Cache | KV Cache Size | Qasper | MF-en | HotpotQA | 2WikiMQA | Musique | GovReport | MultiNews | PRe | LCC | RB-P | Average |
|---|---|---|---|---|---|---|---|---|---|---|---|---|---|
| LLaMA-2-7B-Chat | 16-bit | 1.00× | 19.5 | 34.0 | 30.1 | 26.5 | 10.0 | 24.1 | 26.4 | 9.0 | 59.7 | 53.1 | 29.2 |
| | 2-bit ($g=32$) | 0.22× | 19.0 | 31.4 | 29.7 | 25.1 | 9.8 | 22.0 | 25.3 | 7.0 | 58.8 | 52.8 | 28.1 |
| | + SPECACHE | 0.22× | 20.2 | 32.0 | 30.5 | 26.4 | 10.3 | 24.0 | 26.1 | 8.5 | 59.2 | 53.7 | **29.1** (↑ 1.0) |
| | 2-bit ($g=64$) | 0.19× | 18.4 | 31.1 | 28.7 | 26.6 | 9.8 | 19.9 | 25.3 | 8.5 | 55.8 | 51.1 | 27.5 |
| | + SPECACHE | 0.19× | 19.8 | 32.9 | 30.1 | 27.4 | 9.5 | 23.6 | 25.6 | 9.0 | 59.3 | 53.4 | **29.1** (↑ 1.6) |
| LLaMA-2-13B-Chat | 16-bit | 1.00× | 24.1 | 37.0 | 36.4 | 31.9 | 15.8 | 24.5 | 25.7 | 12.0 | 50.2 | 50.6 | 30.8 |
| | 2-bit ($g=32$) | 0.22× | 23.7 | 36.1 | 35.3 | 31.7 | 14.8 | 22.0 | 25.5 | 12.0 | 49.7 | 48.7 | 30.0 |
| | + SPECACHE | 0.22× | 24.5 | 36.8 | 35.8 | 32.1 | 15.0 | 24.1 | 26.1 | 12.0 | 49.5 | 50.7 | **30.7** (↑ 0.7) |
| | 2-bit ($g=64$) | 0.19× | 22.4 | 34.1 | 36.1 | 32.8 | 14.5 | 21.2 | 25.3 | 12.3 | 49.8 | 48.1 | 29.7 |
| | + SPECACHE | 0.19× | 23.6 | 36.5 | 35.5 | 32.0 | 14.7 | 23.8 | 25.6 | 11.1 | 49.9 | 49.5 | **30.2** (↑ 0.5) |
| Mistral-7B-Instruct-v0.2 | 16-bit | 1.00× | 33.1 | 49.2 | 43.0 | 27.3 | 18.8 | 32.9 | 27.0 | 87.0 | 53.5 | 51.4 | 42.3 |
| | 2-bit ($g=32$) | 0.19× | 31.4 | 49.0 | 42.0 | 26.2 | 18.2 | 32.3 | 26.8 | 76.8 | 52.9 | 51.2 | 40.7 |
| | + SPECACHE | 0.19× | 32.5 | 49.3 | 42.7 | 27.9 | 18.4 | 32.3 | 26.7 | 82.1 | 53.2 | 50.4 | **41.5** (↑ 0.8) |
| | 2-bit ($g=64$) | 0.16× | 31.6 | 47.5 | 41.9 | 28.0 | 18.8 | 31.5 | 26.6 | 68.6 | 52.1 | 50.6 | 39.7 |
| | + SPECACHE | 0.16× | 33.0 | 49.1 | 43.6 | 28.2 | 18.4 | 32.1 | 27.0 | 84.6 | 53.2 | 50.5 | **42.0** (↑ 2.3) |
| | 1-bit ($g=32$) | 0.13× | 18.9 | 40.3 | 32.9 | 25.4 | 13.5 | 20.6 | 22.3 | 41.1 | 46.6 | 42.8 | 30.4 |
| | + SPECACHE | 0.13× | 31.5 | 50.0 | 44.1 | 25.7 | 18.2 | 27.6 | 26.6 | 78.6 | 52.0 | 49.6 | **40.4** (↑ 10.0) |
| | 1-bit ($g=64$) | 0.10× | 18.1 | 38.8 | 31.7 | 23.1 | 13.0 | 21.3 | 22.7 | 34.0 | 44.8 | 41.5 | 28.9 |
| | + SPECACHE | 0.10× | 31.1 | 49.6 | 43.9 | 26.9 | 18.3 | 27.8 | 26.7 | 76.8 | 52.3 | 50.4 | **40.4** (↑ 11.5) |
| LLaMA-3-8B-Instruct | 16-bit | 1.00× | 44.3 | 44.4 | 46.6 | 37.0 | 21.5 | 30.0 | 27.7 | 67.0 | 57.1 | 51.4 | 42.7 |
| | 2-bit ($g=32$) | 0.20× | 43.2 | 44.5 | 46.9 | 37.6 | 20.7 | 29.6 | 27.3 | 67.5 | 51.1 | 47.0 | 41.5 |
| | + SPECACHE | 0.20× | 44.0 | 44.7 | 46.7 | 36.8 | 21.5 | 29.7 | 27.6 | 67.0 | 56.7 | 50.2 | **42.5** (↑ 1.0) |
| | 2-bit ($g=64$) | 0.17× | 43.4 | 44.2 | 46.0 | 36.5 | 20.6 | 29.4 | 27.2 | 65.0 | 53.0 | 50.9 | 41.6 |
| | + SPECACHE | 0.17× | 43.7 | 45.1 | 46.4 | 37.0 | 21.0 | 29.7 | 27.7 | 67.0 | 57.0 | 50.4 | **42.5** (↑ 0.9) |
| | 1-bit ($g=32$) | 0.14× | 26.0 | 32.4 | 40.9 | 28.9 | 16.9 | 7.8 | 6.1 | 64.0 | 18.6 | 25.3 | 26.7 |
| | + SPECACHE | 0.14× | 41.8 | 44.6 | 46.4 | 37.2 | 21.4 | 27.0 | 26.6 | 63.5 | 57.1 | 52.2 | **41.8** (↑ 15.1) |
| | 1-bit ($g=64$) | 0.11× | 23.9 | 29.6 | 39.8 | 25.5 | 15.7 | 4.9 | 5.5 | 62.3 | 17.8 | 22.0 | 24.7 |
| | + SPECACHE | 0.11× | 43.2 | 45.6 | 46.2 | 36.9 | 20.7 | 27.9 | 27.1 | 65.0 | 55.0 | 51.0 | **41.9** (↑ 17.2) |

with index $\mathcal{K}_t$) with 16-bit $\boldsymbol{C}_{\mathcal{K}_t}$ in this computation.

During the attention computation, we also record the indexes $\mathcal{K}_{t+1}$ of top-$k$ KV pairs based on the attention scores of $T'_{t+1}$, which are likely to be needed for decoding $T_{t+1}$ in the next step. Once the computation concludes, we immediately begin prefetching those 16-bit KV pairs $\boldsymbol{C}_{\mathcal{K}_{t+1}}$, evicting the non-top-$k$ 16-bit KV pairs from VRAM that has been used, and offloading the newly created KV pairs. All these memory operations are in parallel with the computation of subsequent layers.

After decoding, $T_t$ and $T'_{t+1}$ generate two tokens: $T_{t+1}$ and $T'_{t+2}$. Since the 16-bit KV pairs needed by $T_t$ were prefetched before its attention computation, we can assume that $T_{t+1}$ is accurate and use it as the model's output. In contrast, $T'_{t+2}$ serves as the speculative token for the next decoding step, since it may be less accurate for output.

The entire inference process of SPECACHE, compared to the original inference process, adds only a single pre-decoding step and decodes two tokens simultaneously during decoding. The additional pre-decoding step is negligible in the overall sentence generation. While the number of decoded tokens increases, the model weights and KV cache used for both tokens are identical. Since the decoding process of LLMs is memory-IO bound, decoding two tokens simultaneously allows shared access to model weights and KV cache, without introducing additional latency. Moreover, as the prefetching runs in parallel with GPU operations, the overall inference latency does not significantly increase.

### 3.4. Post-Training Quantization for GPU KV Cache

Since SPECACHE stores a low-bit copy of the KV cache in the GPU, it can be combined with any KV cache quantization method. Existing post-training KV cache quantization methods, such as KIVI (Liu et al., 2024b), can quantize the KV cache to 2-bit without calibration, while methods like

*Table 2.* **Comparison with other methods on LongBench.** SPECACHE outperforms the other methods in terms of average performance, even when using higher compression ratios.

| | Method | KV Cache Size | Qasper | MF-en | HotpotQA | 2WikiMQA | Musique | GovReport | MultiNews | PRe | LCC | RB-P | Average |
|---|---|---|---|---|---|---|---|---|---|---|---|---|---|
| Mistral-7B-Instruct-v0.2 | Full KV cache | 1.00× | 33.1 | 49.2 | 43.0 | 27.3 | 18.8 | 32.9 | 27.0 | 87.0 | 53.5 | 51.4 | 42.3 |
| | InfLLM | 0.25× | 16.8 | 38.4 | 33.9 | 19.2 | 18.2 | 29.6 | 24.7 | 41.4 | 52.8 | 55.3 | 33.0 |
| | StreamLLM | 0.25× | 15.1 | 25.4 | 27.7 | 17.4 | 14.6 | 27.4 | 22.1 | 31.6 | 51.8 | 55.9 | 28.9 |
| | H$_2$O | 0.25× | 24.9 | 44.8 | 35.0 | 19.0 | 17.5 | 28.6 | 24.2 | 82.9 | 53.9 | 52.3 | 38.3 |
| | SPECACHE | **0.16×** | 33.0 | 49.1 | 43.6 | 28.2 | 18.4 | 32.1 | 27.0 | 84.6 | 53.2 | 50.5 | **42.0** |
| | InfLLM | 0.13× | 12.8 | 33.0 | 29.1 | 16.2 | 13.3 | 27.9 | 23.8 | 26.2 | 54.1 | 53.5 | 29.0 |
| | StreamLLM | 0.13× | 11.3 | 22.9 | 22.8 | 12.0 | 10.7 | 24.6 | 19.7 | 16.9 | 53.8 | 56.0 | 25.1 |
| | H$_2$O | 0.13× | 21.4 | 41.1 | 32.8 | 16.8 | 15.9 | 26.2 | 23.0 | 79.5 | 52.8 | 51.8 | 36.1 |
| | SPECACHE | **0.10×** | 31.1 | 49.6 | 43.9 | 26.9 | 18.3 | 27.8 | 26.7 | 76.8 | 52.3 | 50.4 | **40.4** |
| LLaMA-3-8B-Instruct | Full KV Cache | 1.00× | 44.3 | 44.4 | 46.6 | 37.0 | 21.5 | 30.0 | 27.7 | 67.0 | 57.1 | 51.4 | 42.7 |
| | InfLLM | 0.25× | 28.0 | 35.1 | 36.6 | 23.0 | 14.4 | 29.6 | 25.5 | 37.5 | 56.9 | 58.3 | 34.5 |
| | StreamLLM | 0.25× | 23.0 | 21.1 | 29.8 | 24.7 | 12.0 | 25.9 | 22.6 | 21.0 | 55.0 | 57.2 | 29.2 |
| | H$_2$O | 0.25× | 41.2 | 41.8 | 46.8 | 36.9 | 21.5 | 25.7 | 23.7 | 66.0 | 55.1 | 51.2 | 41.0 |
| | SPECACHE | **0.17×** | 43.7 | 45.1 | 46.4 | 37.0 | 21.0 | 29.7 | 27.7 | 67.0 | 57.0 | 50.4 | **42.5** |
| | InfLLM | 0.13× | 22.0 | 27.0 | 33.1 | 20.5 | 9.3 | 27.5 | 24.4 | 18.0 | 61.4 | 58.4 | 30.2 |
| | StreamLLM | 0.13× | 16.6 | 17.4 | 25.7 | 18.5 | 9.8 | 23.4 | 19.9 | 8.0 | 60.4 | 55.8 | 25.6 |
| | H$_2$O | 0.13× | 37.8 | 42.1 | 46.6 | 36.9 | 21.5 | 23.7 | 21.9 | 65.5 | 54.6 | 50.8 | 40.1 |
| | SPECACHE | **0.11×** | 43.2 | 45.6 | 46.2 | 36.9 | 20.7 | 27.9 | 27.1 | 65.0 | 55.0 | 51.0 | **41.9** |

KVQuant (Hooper et al., 2024) achieve 1-bit quantization with the help of calibration. We utilize KIVI to quantize the GPU copy due to its simplicity and the fact that it does not require calibration. To further push the limits of KV cache compression, we modify KIVI for 1-bit quantization.

The original KIVI quantizes KV cache as follows:

$$Q(X) = \lfloor \frac{X - z_X}{s_X} \rceil, \quad X' = Q(X) \cdot s_X + z_X, \quad (2)$$

where $z_X = \min X$ is the zero-point, $s_X = (\max X - \min X)/(2^B - 1)$ is the scaling factor, and $\lfloor \cdot \rceil$ is the rounding operation.

However, in 1-bit quantization, the elements of $X'$ are either $\max X$ or $\min X$. This results in the quantized KV cache having an unusually large magnitude, which makes the model struggle to perform text generation effectively. To address this issue, we improve KIVI in the 1-bit scenario by assuming that the weights follow a uniform distribution between the minimum $\min X$ and $\max X$, and ensuring that the cumulative quantization error is minimized. Based on this, the zero-point and scaling factor are modified as follows:

$$z_X = \frac{3 \cdot \min X + \max X}{4}, s_X = \frac{\max X - \min X}{2} \quad (3)$$

which ensures that all values in the range $[\min X, (\min X + \max X)/2)$ are quantized to the midpoint of this interval,

$(3 \cdot \min X + \max X)/4$, and values in the range $[(\min X + \max X)/2, \max X]$ are quantized to the midpoint of this interval, $\min X + 3 \cdot \max X)/4$.

## 4. Experiments

### 4.1. Implementation Details

Note that KIVI retains a small set of recent KV pairs as residuals to perform channel-wise quantization of the keys and enhance model performance. Since SPECACHE temporarily loads a small number of KV pairs into the GPU, for a fair comparison, we reduce the number of residual KV pairs used by SPECACHE, ensuring that SPECACHE does not increase VRAM usage on top of KIVI. Specifically, for the baseline KIVI method, we use 128 residual KV pairs and quantization group size of 32 and 64. For SPECACHE, we prefetch top-64 16-bit KV pairs from CPU memory. Since these 16-bit KV pairs will be loaded into VRAM, in order to ensure that the total size of the KV cache remains unchanged, we use a smaller residual length of 64.

### 4.2. Performance on LongBench

For LongBench, we report results on 10 tasks: Qasper, MultiFieldQA, HotpotQA, 2WikiMQA, MuSiQue, GovReport, MultiNews, PassageRetrieval, LCC, and RepoBench-P. Full results on all the 15 tasks are provided in Appendix. The maximum sequence length is set to 4k for LLaMA-2, 32k

*Table 3.* **Decoding throughput.** Evaluated on a single NVIDIA A6000 GPU using Mistral-7B-Instruct-v0.2.

| Context | Full KV cache | | SPECACHE (2-bit, $g = 64$) | | | SPECACHE (1-bit, $g = 64$) | | |
|---|---|---|---|---|---|---|---|---|
| | Batch Size | Throughput | Batch Size | Throughput | Speedup | Batch Size | Throughput | Speedup |
| 2k | 44 | 190.3 | 272 | 418.5 | 2.2× | 410 | 526.3 | 2.8× |
| 8k | 11 | 47.0 | 82 | 133.7 | 2.8× | 134 | 177.7 | 3.8× |
| 32k | 3 | 10.3 | 22 | 34.6 | 3.4× | 36 | 47.3 | 4.6× |

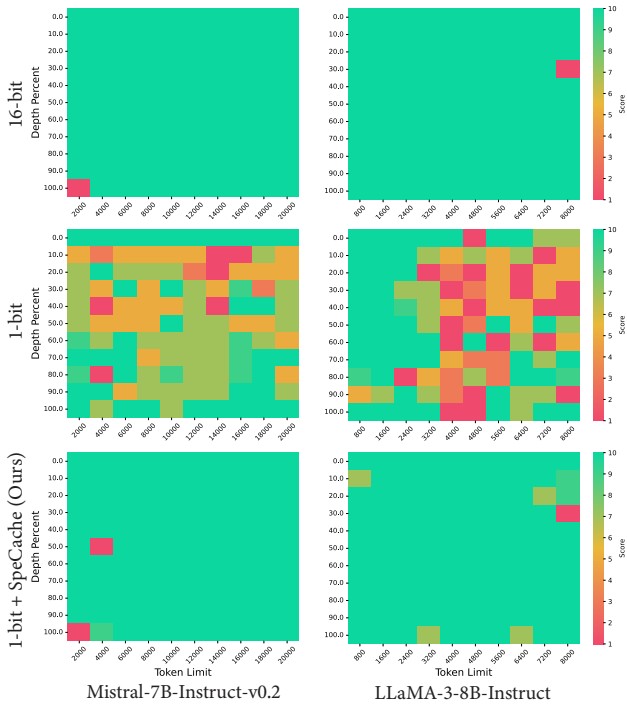

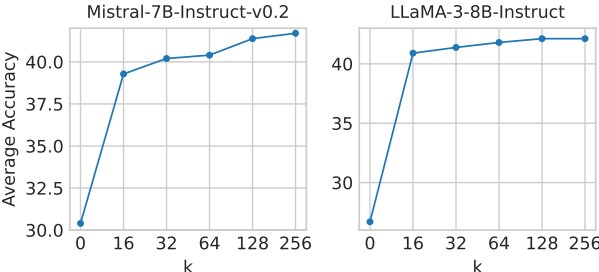

*Figure 5.* **Ablation study on** $k$**.** We use 1-bit SPECACHE with $g = 32$.

In addition to KIVI, we also compare SPECACHE with other representative KV cache compression methods, including InfLLM (Xiao et al.), H2O (Zhang et al., 2023), and Stream-LLM (Xiao et al., 2024). For these baselines, we follow the settings from Yuan et al. (2024), compressing the KV cache to 1/4 or 1/8 of its original size, and compare them with 2-bit and 1-bit SPECACHE, respectively. The residual length and group size for SPECACHE are set to 64. As shown in Table 2, SPECACHE outperforms the other methods in terms of average performance, even when using higher compression ratios.

### 4.3. Performance on Needle-in-a-Haystack Benchmark

First, we evaluate the model's long-context retrieval ability after applying KV cache compression with SPECACHE on a synthetic task — the Needle-in-a-Haystack (NIAH) benchmark (Greg Kamradt, 2023). We report retrieval accuracy under three different settings on the Mistral-7B-Instruct-v0.2 model with a 32k context and the LLaMA-3-8B-Instruct model with an 8k context: 16-bit KV cache, where no KV cache compression is applied; 1-bit KV cache, where we use KIVI to quantizes the KV cache to 1-bit; and 1-bit KV cache + SpeCache, where SpeCache is used to prefetch the 16-bit KV cache from CPU memory on top of 1-bit quantization.

The results are shown in Figure 4. Under 1-bit KV cache quantization, LLMs' long-context retrieval ability is significantly compromised. After applying SPECACHE, the performance of LLMs under 1-bit KV cache quantization is comparable to that of the 16-bit.

*Figure 4.* **Performance on Needle-in-a-haystack benchmark.** We use $g = 32$ for quantization, resulting in the KV cache compression ratio of 0.13 and 0.14 for Mistral-7B-Instruct-v0.2 and LLaMA-3-8B-Instruct, respectively.

for Mistral, and 8k for LLaMA-3. For LLaMA-2, we only opt for 2-bit KV cache quantization, as the model's performance under 1-bit KV cache quantization is significantly poor.

According to the results are shown in Table 1, we have several observations. As the degree of KV compression increases (with lower bit widths or larger group sizes), the model's performance declines. However, when combined with SPECACHE, the majority of the performance loss is recovered. We find that the greater the KV compression, the more pronounce the advantages of our method become, which further demonstrates that top-$k$ KV cache can recover most of the attention information. Specifically, our method can maintain a performance gap of only 2% and 1% compared to the 16-bit baseline for Mistral-7B-Instruct-v0.2 and LLaMA-3-8B-Instruct, respectively, while retaining only approximately 10% of the KV cache size in the GPU.

*Table 4.* **Comparison with non-speculative fetching.** We report the average decoding latency for a single step when using the maximum batch size that 48GB VRAM of a single NVIDIA A6000 can handle with a context length of 2k.

| W/ Spec. Prefetch | Bit-width | Qasper | MF-en | HotpotQA | 2WikiMQA | Musique | GovReport | MultiNews | PRe | LCC | RB-P | Average | Latency (ms/step) |
|---|---|---|---|---|---|---|---|---|---|---|---|---|---|
| ✗ | 2-bit | 44.5 | 44.9 | 46.4 | 37.4 | 21.4 | 29.6 | 27.8 | 66.5 | 57.3 | 50.5 | 42.6 | 877 |
| ✓ | 2-bit | 43.7 | 45.1 | 46.4 | 37.0 | 21.0 | 29.7 | 27.7 | 67.0 | 57.0 | 50.4 | 42.5 | **643** |
| ✗ | 1-bit | 43.3 | 44.7 | 46.7 | 37.2 | 21.8 | 28.9 | 27.5 | 66.0 | 58.3 | 50.1 | 42.4 | 1144 |
| ✓ | 1-bit | 43.2 | 45.6 | 46.2 | 36.9 | 20.7 | 27.9 | 27.1 | 65.0 | 55.0 | 51.0 | 41.9 | **775** |

*Table 5.* **Ablation study on quantization method.** Tested on LLaMA-3-8B-Instruct.

| Improved 1-bit KIVI | SPECACHE | Qasper | 2WikiMQA | RB-P |
|---|---|---|---|---|
| ✗ | ✗ | 3.7 | 4.9 | 20.3 |
| ✗ | ✓ | 4.1 | 6.5 | 21.6 |
| ✓ | ✗ | 23.9 | 25.5 | 22.0 |
| ✓ | ✓ | **43.2** | **36.9** | **51.0** |

### 4.4. Efficiency

To highlight the advantages of SPECACHE in reducing KV cache size, we test its maximum throughput during decoding. We conduct tests in scenarios with context lengths of 2k, 8k, and 32k, increasing the batch size to maximize GPU memory usage up to the 48GB VRAM of an NVIDIA A6000 GPU. The decoding throughput measured on Mistral-7B-Instruct-v0.2 model with HuggingFace `transformers` as framework. Note that our CPU-GPU interaction code is implemented using `pytorch`'s multi-stream mechanism and the `Tensor.copy_()` method, so the parallelism achieved is not theoretically optimal. By customizing lower-level operators, the efficiency of SPECACHE can be further improved.

As shown in the Table 3, with 2-bit and 1-bit quantization, SPECACHE allows the batch size to increase by up to 7× and 12×, respectively, resulting in overall throughput improvements of 3.4× and 4.6× compared to the original setup. Notably, when the context length is large, the original KV cache can only accommodate smaller batch sizes, leading to lower parallelism. This results in more significant acceleration with SPECACHE.

### 4.5. Ablation Experiments

**Ablation study on $k$.** We conduct an ablation study on the number of prefetched KV pairs, $k$. As shown in the Figure 5, even with a small $k$, such as $k = 16$, SPECACHE still provides a significant improvement over the KIVI baseline (*i.e.*, $k = 0$). As $k$ increases, the model's performance improves, but at the same time, a larger $k$ increases the size

of transferred data. In practical applications, a trade-off for $k$ can be made.

**Comparison with original 1-bit KIVI.** To highlight the importance of the new zero point and scaling factor we propose for 1-bit KIVI, we conduct an ablation study. As shown in Table 5, when using the original KIVI, although SPECACHE provides some improvements, the model's performance is still very poor, regardless of whether SPECACHE is used. After modifying the quantization method, the model's performance significantly improves, and the advantages of SPECACHE become much more apparent.

**Comparison with non-speculative fetching.** One advantage of our method is the ability to prefetch KV pairs one step ahead, allowing prefetching and computation to run in parallel. We compare this with another simple strategy, where instead of using speculative tokens, we use the output tokens from the previous step to estimate the top-$k$ KV pairs and fetch them, followed by recalculating attention. In this strategy, the fetching starts after the attention computation is completed and needs to be finished before recalculating attention, meaning it cannot run in parallel.

As shown in Table 4, using accurate output tokens instead of speculative tokens slightly improves the model's performance, but at the cost of significantly increasing the model's latency. As the batch size increases, the time required to load KV pairs from the CPU increases linearly, while the computation time increases more slowly due to the improved parallelism. Therefore, during decoding with large batch sizes, the latency of loading KV pairs becomes the dominant factor. This highlights the importance of parallelizing computation and prefetching.

## 5. Conclusion

In this paper, we propose SPECACHE, a low-latency, high-performance, and training-free KV cache compression method. SPECACHE stores high-precision KV cache in CPU RAM, while low-bit (down to 1 bit) KV cache is stored in GPU VRAM. It utilizes speculative tokens and output tokens co-decoding to anticipate and prefetch the

top-$k$ KV pairs for the next decoding step. This approach effectively recovers the information lost due to the low-bit KV cache while enabling the parallelization of prefetching and computation.

## Impact Statement

This paper is based on LLMs, which have potential societal impacts, including concerns around bias, misinformation, and accessibility. While these issues are well-known, we aim to contribute to improving the efficiency of LLMs. Continued ethical oversight and interdisciplinary collaboration are essential as the field evolves.

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

# Appendix

## A. Algorithm

We provide the pseudocode for a single attention layer. For simplicity, we have omitted the residual and grouped quantization details of KIVI.

---
**Algorithm 1** Prefilling
---
    **Input:** $T \in \mathbb{R}^{L \times d}$
    $Q = TW_q, K = TW_k, V = TW_v$
    $K' = Quant(K), V' = Quant(V)$
    $C = \{V, K\}, C' = \{V', K'\}$
    $Offload(C)$
    $A = MaskedSoftmax(QK^\top)$
    $O = AVW_o$
    **Output:** $O \in \mathbb{R}^{L \times d}$
---

---
**Algorithm 2** Pre-decoding
---
    **Input:** $T_1 \in \mathbb{R}^{1 \times d}$
    $Q_1 = T_1 W_q, K_1 = T_1 W_k, V_1 = T_1 W_v$
    $K = [K', K_1], V = [V', V_1]$
    $A = Softmax(Q_1 K^\top)$
    $\mathcal{K}_1 = ArgTopK(A)$
    $Pre\text{-}fetch(C_{\mathcal{K}_1})$
    $O = AVW_o$
    **Output:** $O \in \mathbb{R}^{1 \times d}$
---

---
**Algorithm 3** Decoding
---
    **Input:** $T = [T_t, T'_{t+1}] \in \mathbb{R}^{2 \times d}$
    $Q_t = TW_q, K_t = TW_k, V_t = TW_v$
    $K'_t = Quant(K_t), V'_t = Quant(V_t)$
    $K = [K' \cup K_{\mathcal{K}_t}, K_t], V = [V' \cup V_{\mathcal{K}_t}, V_t]$
    $A = MaskedSoftmax(Q_t K^\top)$
    $\mathcal{K}_{t+1} = ArgTopK(A_{1,:})$
    $Pre\text{-}fetch(C_{\mathcal{K}_{t+1}})$
    $C_t = \{V_t, K_t\}, C'_t = \{V'_t, K'_t\}$
    $C = [C, Offload(C_t)], C' = [C', C'_t]$
    $O = AVW_o$
    **Output:** $O \in \mathbb{R}^{2 \times d}$
---

## B. Setting of Needle-in-a-Haystack Benchmark

We notice that the NIAH benchmark has many different versions. We follow the setting from Greg Kamradt (2023), inserting the sentence "*The best thing to do in San Francisco is eat a switch and sit in Dolores Park on a sunny day* into Paul Graham's essays, and added the question, "*What is the best thing to do in San Francisco? Here is the most relevant sentence in the context:*" After that, we use GPT-4 to score the model's response based on the following criteria:

- Score 1: The answer is completely unrelated to the reference.

- Score 3: The answer has minor relevance but does not align with the reference.

- Score 5: The answer has moderate relevance but contains inaccuracies.

- Score 7: The answer aligns with the reference but has minor omissions.

- Score 10: The answer is completely accurate and aligns perfectly with the reference.

## C. Full Results on LongBench

Due to space constraints, we only report the results for ten tasks from LongBench in the main text. The results for all 15 tasks are shown in the table below. Notably, both KIVI and SPECACHE use 2-bit and 1-bit quantization with $g = 64$.

| | Method | KV Cache Size | NarrativeQA | Qasper | MF-en | HotpotQA | 2WikiMQA | Musique | GovReport | QMSum | MultiNews | TREC | TriviaQA | SAMSum | PRe | LCC | RB-P | Average |
|---|---|---|---|---|---|---|---|---|---|---|---|---|---|---|---|---|---|---|
| **Mistral-7B-Instruct-v0.2** | Full KV cache | 1.00× | 26.9 | 33.1 | 49.2 | 43.0 | 27.3 | 18.8 | 32.9 | 24.2 | 27.0 | 71.0 | 86.2 | 42.8 | 87.0 | 53.5 | 51.4 | 45.0 |
| | InfLLM | 0.25× | 20.9 | 16.8 | 38.4 | 33.9 | 19.2 | 18.2 | 29.6 | 22.2 | 24.7 | 60.5 | 88.3 | 41.3 | 41.4 | 52.8 | 55.3 | 37.5 |
| | StreamLLM | 0.25× | 19.7 | 15.1 | 25.4 | 27.7 | 17.4 | 14.6 | 27.4 | 20.2 | 22.1 | 61.0 | 83.7 | 39.2 | 31.6 | 51.8 | 55.9 | 34.2 |
| | H$_2$O | 0.25× | 21.6 | 24.9 | 44.8 | 35.0 | 19.0 | 17.5 | 28.6 | 22.8 | 24.2 | 71.0 | 86.7 | 43.8 | 82.9 | 53.9 | 52.3 | 41.9 |
| | KIVI (2-bit) | 0.16× | 27.0 | 31.6 | 47.5 | 41.9 | 28.0 | 18.8 | 31.5 | 23.6 | 26.6 | 71.0 | 86.2 | 43.6 | 68.6 | 52.1 | 50.6 | 43.2 |
| | SPECACHE | **0.16×** | 27.2 | 33.0 | 49.1 | 43.6 | 28.2 | 18.4 | 32.1 | 27.1 | 27.1 | 71.0 | 85.9 | 42.8 | 84.6 | 53.2 | 50.5 | **44.9** |
| | InfLLM | 0.13× | 20.9 | 12.8 | 33.0 | 29.1 | 16.2 | 13.3 | 27.9 | 21.2 | 23.8 | 60.0 | 86.0 | 40.1 | 26.2 | 54.1 | 53.5 | 34.5 |
| | StreamLLM | 0.13× | 16.8 | 11.3 | 22.9 | 22.8 | 12.0 | 10.7 | 24.6 | 19.8 | 19.7 | 56.5 | 79.6 | 38.8 | 16.9 | 53.8 | 56.0 | 30.8 |
| | H$_2$O | 0.13× | 20.9 | 21.4 | 41.1 | 32.8 | 16.8 | 15.9 | 26.2 | 22.6 | 23.0 | 71.0 | 86.3 | 43.8 | 79.5 | 52.8 | 51.8 | 40.4 |
| | KIVI (1-bit) | **0.10×** | 21.7 | 18.1 | 38.8 | 31.7 | 23.1 | 13.0 | 21.3 | 20.8 | 22.7 | 41.0 | 77.8 | 38.4 | 34.0 | 44.8 | 41.5 | 32.6 |
| | SPECACHE | **0.10×** | 27.2 | 31.1 | 49.6 | 43.9 | 26.9 | 18.3 | 27.8 | 24.0 | 26.7 | 70.5 | 86.6 | 41.6 | 76.8 | 52.3 | 50.4 | **43.6** |
| **LLaMA-3-8B-Instruct** | Full KV Cache | 1.00× | 21.7 | 44.3 | 44.4 | 46.6 | 37.0 | 21.5 | 30.0 | 22.6 | 27.7 | 74.5 | 90.6 | 42.7 | 67.0 | 57.1 | 51.4 | 45.3 |
| | InfLLM | 0.25× | 18.1 | 28.0 | 35.1 | 36.6 | 23.0 | 14.4 | 29.6 | 19.8 | 25.5 | 61.0 | 88.8 | 42.0 | 37.5 | 56.9 | 58.3 | 38.3 |
| | StreamLLM | 0.25× | 17.4 | 23.0 | 21.1 | 29.8 | 24.7 | 12.0 | 25.9 | 19.5 | 22.6 | 60.5 | 85.7 | 40.5 | 21.0 | 55.0 | 57.2 | 34.4 |
| | H$_2$O | 0.25× | 21.8 | 41.2 | 41.8 | 46.8 | 36.9 | 21.5 | 25.7 | 21.4 | 23.7 | 74.0 | 90.6 | 42.4 | 66.0 | 55.1 | 51.2 | 44.0 |
| | KIVI (2-bit) | 0.17× | 21.0 | 43.4 | 44.2 | 46.0 | 36.5 | 20.6 | 29.4 | 21.9 | 27.2 | 74.0 | 90.1 | 42.9 | 65.0 | 53.0 | 50.9 | 44.4 |
| | SPECACHE | **0.17×** | 20.9 | 43.7 | 45.1 | 46.4 | 37.0 | 21.0 | 29.7 | 22.7 | 27.7 | 74.5 | 90.1 | 41.9 | 67.0 | 57.0 | 50.4 | **45.0** |
| | InfLLM | 0.13× | 14.2 | 22.0 | 27.0 | 33.1 | 20.5 | 9.3 | 27.5 | 19.1 | 24.4 | 58.0 | 82.0 | 40.9 | 18.0 | 61.4 | 58.4 | 34.4 |
| | StreamLLM | 0.13× | 13.1 | 16.6 | 17.4 | 25.7 | 18.5 | 9.8 | 23.4 | 18.2 | 19.9 | 55.5 | 72.3 | 39.9 | 8.0 | 60.4 | 55.8 | 30.3 |
| | H$_2$O | 0.13× | 21.3 | 37.8 | 42.1 | 46.6 | 36.9 | 21.5 | 23.7 | 21.1 | 21.9 | 74.0 | 90.5 | 42.9 | 65.5 | 54.6 | 50.8 | 43.4 |
| | KIVI (1-bit) | **0.11×** | 18.6 | 23.9 | 29.6 | 39.8 | 25.5 | 15.7 | 4.9 | 17.5 | 5.5 | 66.0 | 67.1 | 34.8 | 62.3 | 17.8 | 22.0 | 30.1 |
| | SPECACHE | **0.11×** | 21.0 | 43.2 | 45.6 | 46.2 | 36.9 | 20.7 | 27.9 | 21.3 | 27.1 | 74.0 | 90.4 | 40.1 | 65.0 | 55.0 | 51.0 | **44.4** |

