# OpenReview forum: "SpeCache: Speculative Key-Value Caching for Efficient Generation of LLMs"
_ICML.cc/2025/Conference — ICML 2025 poster_

### Official Review · Reviewer_xLPX · 2025-03-12

**Overall Recommendation:** 3

**Summary:**

The paper introduces SpeCache, a speculative KV caching mechanism designed to enhance the efficiency of LLM inference. SpeCache mitigates these drawbacks by offloading the complete KV cache to CPU memory and dynamically fetching KV pairs back into GPU memory during decoding. To minimize CPU-GPU transfer latency, it employs a speculative mechanism to predict the KV pairs required for the next decoding step, allowing memory transfer and computation to proceed in parallel. The method does not require retraining and significantly reduces memory usage while maintaining competitive accuracy.

## update after rebuttal

The authors’ response partially addresses my concerns. However, based on the discussions raised by other reviewers and overall rebuttal, I believe that my key concerns remain unresolved. Below, I elaborate on the remaining issues:

While SpeCache proposes a KV cache offloading technique to address the long-context problem, the motivation described in the paper does not adequately support this objective. Since the method is based on offloading KV cache from the GPU to CPU memory (not merely loading from GPU DRAM), the examples given in the introduction and the maximum sequence lengths of the models evaluated in the experimental section do not convincingly demonstrate the utility of SpeCache in genuinely resource-constrained environments. As such, I find it difficult to accept the claim that SpeCache is effective under constrained memory settings.

This paper presents Table 1 and Table 2 as the main evaluation results, but there are still concerns about whether these comparisons are fair and appropriate.  In Table 1, the comparison with KIVI fixes the residual length at 64 and allocates 16-bit precision to the top-64 KV pairs. It is intuitive that assigning higher bit-precision to more important tokens would yield higher accuracy, making the comparison less informative. In Table 2, SpeCache is compared against baselines that adopt eviction strategies, while SpeCache itself stores the full KV cache in CPU memory (non-eviction). This setup favors SpeCache by design and does not ensure a fairness in accuracy comparisons.

The throughput evaluation also has room for improvement. The reported throughput gains are primarily achieved by increasing the batch size, which is similar to the benefits observed in prior work through KV cache size reduction. Moreover, the comparison is only made against FullKV, without including other CPU-GPU offloading methods. A more meaningful evaluation would involve direct comparisons with other CPU-GPU offloading methods, including the overhead from KV cache dequantization and other related costs.

While I appreciate the authors’ additional efforts in providing further experimental results, the fundamental concerns I raised have not been fully resolved. Therefore, I will maintain my original score.

**Claims And Evidence:**

The claims in the paper are generally supported by clear evidence, but there are some points that deserve closer examination. The following is an evaluation of the claims and the evidence provided in this paper:

**1. Unclear Overhead of Parallel Prefetching and Computation**

•	The paper states that "prefetching and computation can occur in parallel, avoiding any increase in inference latency." However, while Table 4 presents the latency gain when fetching and computation are executed sequentially, it does not provide results on the additional overhead introduced by the parallel prefetching and computation framework compared to standard inference.

•	Without explicit analysis of the potential trade-offs, it is unclear how much overhead the parallel execution introduces in real-world inference scenarios, particularly when considering factors such as kernel launch overhead, synchronization penalties, or impact on GPU utilization.


**2. Questionable Relevance of the KV Cache Bottleneck Example**

•	The introduction section states that for LLaMA-7B with a batch size of 16 and sequence length of 2k, the KV cache size reaches 8.4B parameters, and that this can be a bottleneck in memory-constrained environments like on-device inference.

•	However, in local on-device inference, single-batch inference is more common than large batch sizes. A more appropriate example would be a single-batch input with a sequence length of 128k, which better reflects real-world usage scenarios.

•	Recent models such as LLaMA-3.1-8B already support 128k sequence length with GQA, making the assumption that 2k is a long context less relevant. Furthermore, for a single batch with a 2k input length, the KV cache size is only about 0.26GB, which is unlikely to be a severe memory constraint in modern hardware setups. This weakens the claim that a 2k length input presents a significant long-context challenge.

**3. Peak Memory Usage Concern in Prefill Phase**

•	One limitation of SpeCache is that during the prefill phase, the entire KV cache must be deployed on the GPU at least once before any offloading occurs.

•	This means that peak memory consumption is not reduced, which can be a critical constraint for on-device inference applications with limited memory.

•	The paper does not discuss whether SpeCache enables inference in environments where the full KV cache would otherwise exceed memory limits, which is crucial for evaluating its feasibility in resource-constrained scenarios.

**Essential References Not Discussed:**

The paper discusses all major prior works related to KV caching, attention sparsity, and speculative execution. No significant omissions were noted.

**Experimental Designs Or Analyses:**

1. The LongBench dataset includes sequences exceeding 32K tokens. However, the evaluation in the paper truncates sequences to 4K, 8K, or 32K depending on the model's maximum context length. This truncation lowers the upper bound of accuracy for full KV cache models, potentially making the reported accuracy gap between full KV cache and compressed KV cache models appear smaller than it would be with longer sequences. A more appropriate evaluation should be conducted using a model that supports 128K sequences without truncation, such as LLaMA-3.1-8B-Instruct, to ensure fair comparisons.

2. The paper evaluates SpeCache against older methods such as H2O and StreamLLM, but does not compare it with more recent and competitive methods like SnapKV [1].

3. Table 3 demonstrates the throughput improvements of SpeCache by increasing the maximum batch size due to KV cache compression. However, this approach does not isolate the direct impact of KV cache compression on KV cache loading time. A better approach would be to compare throughput gains at a fixed batch size, measuring the speed-up factor when using full KV cache versus SpeCache. This would more clearly demonstrate how much KV compression reduces KV cache loading time rather than conflating improvements from increased batch sizes.

[1] Yuhong Li, et al., "SnapKV: LLM Knows What You are Looking for Before Generation", arXiv:2404.14469.

**Methods And Evaluation Criteria:**

The paper introduces a novel speculative KV caching method, integrating CPU offloading with speculative token decoding. The evaluation is based on:

•	Accuracy on LongBench and Needle-in-a-Haystack benchmarks

•	Memory efficiency (compression ratio and VRAM usage)

**Other Comments Or Suggestions:**

Typo in line 302: “min X + 3 max X)” -> “(min X + 3 max X)”

**Other Strengths And Weaknesses:**

**Strength**

1. The paper presents a novel speculative prefetching mechanism for KV cache in LLMs, which differentiates it from conventional KV cache compression and offloading methods.

2. Unlike previous methods that rely on compression techniques such as quantization, merging, or eviction, SpeCache introduces speculative tokens to anticipate the next KV pairs required for decoding, which is an innovative approach.

**Weakness**

1. Unclear assumption for long context (See 	Claims and Evidence 1. and 2.)

2. Peak Memory Usage Concern (See Claims and Evidence 3.)

3. Unreasonable experimental results (See Experimental Designs or Analyses)

**Questions For Authors:**

See Claims and Evidence and Experimental Designs and Weakness.

**Relation To Broader Scientific Literature:**

The paper builds on and extends existing work in:

•	KV cache compression

•	Offloading techniques

•	Speculative execution for LLMs

The method is positioned as a training-free enhancement, making it broadly applicable.

**Theoretical Claims:**

The paper does not present a formal theoretical analysis but relies on empirical justification. The main theoretical intuition is that attention sparsity allows selective KV caching without performance loss. This assumption is supported by quantitative studies of attention sparsity and cache hit rates.

---

> ### Author Rebuttal · Authors · 2025-03-31
>
> We thank the reviewer for their thoughtful feedback! We address specific concerns and questions below.
>
>  > Q1. Unclear Overhead of Parallel Prefetching and Computation
>
> Since our method is built upon KIVI, we can consider KIVI as the baseline when parallel pre-fetching and computation are not used. On Mistral-7B-Instruct-v0.2, we test the latency and VRAM usage of KIVI and KIVI+SpeCache with batch size of 1 and context length of 64k.
>
> |Method|Latency (ms/step)|Allocated memory (GB)|
> |-|-|-|
> |Full KV|204.3|31.1|
> |2-bit KIVI (g=64)|94.6|22.8|
> |+ SpeCache|101.6|22.8|
> |1-bit KIVI (g=64)|94.3|22.3|
> |+ SpeCache|103.3|22.3|
>
> In addition, we also provide the throughput of Mistral-7B-Instruct-v0.2 with context length of 32k when the batch size is increased to the maximum capacity that can be accommodated by 48GB.
>
> |Method|Throughput (tok/sec)|batch size|
> |-|-|-|
> |2-bit KIVI (g=64)|36.5|22|
> |+ SpeCache|34.6|22|
> |1-bit KIVI (g=64)|50.8|36|
> |+ SpeCache|47.3|36|
>
> We can find that SpeCache only slightly increase the latency of KIVI.
>
>  > Q2. Questionable Relevance of the KV Cache Bottleneck Example
>
> Thank you for pointing that out. Our example is indeed somewhat outdated. We will revise the description to specify the KV cache size for long sequences with a batch size of 1, such as for Mistral-v0.2-7B with a 32k context length, where the KV cache size exceeds 4B, and for Llama3.1 with a 128k context length, where the KV cache size exceeds 16B.
>
>  > Q3. Peak Memory Usage Concern in Prefill Phase
>
> In fact, during the prefilling phase, **we do not need to keep the entire KV cache in GPU memory**. This is because the computation, quantization, and offloading of the KV cache are done layer-by-layer. In other words, at any given moment, only the KV cache of a single layer needs to be fully stored in GPU memory. Before the KV cache of one layer is computed, the KV cache from the previous layer has already been quantized and offloaded. For example, Mistral-7B has 32 layers, so at any given time, we only need 1/32 of the 16-bit KV cache in GPU memory. This also allows SpeCache to handle context lengths that full cache cannot accommodate.
>
>  > Q4. Experiments on LLaMA-3.1-8B-Instruct
>
>  We implemented SpeCache on Llama-3.1-8B-instruct and evaluated it on longbench.
>
> |Method|KV Size|Qasper|MF-en|HotpotQA|2WikiMQA|Musique|GovReport|MultiNews|PRe|LCC|RB-P|Average|
> |-|-|-|-|-|-|-|-|-|-|-|-|-|
> |16-bit Full KV|1.00x|45.5|54.9|56.0|46.6|31.3|34.6|27.2|99.5|63.2|55.4|51.4|
> |H2O|0.13x|38.8|42.6|43.2|37.7|25.8|26.0|21.2|96.0|54.1|49.5|43.5|
> |1-bit KIVI (g=64)|0.10x|22.9|30.1|38.7|22.0|16.9|11.0|14.9|76.5|39.1|34.1|30.6|
> |+ SpeCache|0.10x|43.0|53.0|50.4|40.6|30.4|34.7|27.5|98.0|57.7|48.1|48.4|
>
> As you expected, the performance of Full KV significantly improves compared to the 8K context at 128K context. Our method, with just 10% KV size, has a 3% gap compared to Full KV, but still shows a 17.8% improvement over the purely quantization method, KIVI.
>
>  > Q5. Comparision with SnapKV
>
> We added SnapKV as a baseline on longbench. Since SnapKV compresses the KV cache to a fixed size, to ensure a fair comparison, we dynamically calculated the SnapKV budget for each sample length, aligning its KV cache size with that of our method.
>
> |Model|Method|KV Size|Qasper|MF-en|HotpotQA|2WikiMQA|Musique|GovReport|MultiNews|PRe|LCC|RB-P|Average|
> |-|-|-|-|-|-|-|-|-|-|-|-|-|-|
> |Mistral-7B-Instruct-v0.2|SnapKV|0.10x|25.0|45.1|34.1|18.7|18.8|26.2|24.4|79.5|51.8|51.4|37.5|
> |Mistral-7B-Instruct-v0.2|1-bit SpeCache (g=64)|0.10x|31.1|49.6|43.9|26.9|18.3|27.8|26.7|76.8|52.3|50.4|40.4|
> |LLaMA-3-8B-Instruct|SnapKV|0.11x|39.1|42.9|46.2|36.1|21.8|23.2|21.5|66.0|55.1|50.2|40.2|
> |LLaMA-3-8B-Instruct|1-bit SpeCache (g=64)|0.11x|43.2|45.6|46.2|36.9|20.7|27.9|27.1|65.0|55.0|51.0|41.9|
>
> Under the same KV cache size budget, SnapKV outperforms all other baselines. However, SpeCache still delivers superior performance. Our method is capable of working with an extremely small KV cache size (about 0.1x), whereas SnapKV tends to experience performance degradation under such conditions. This is because SnapKV compresses the KV cache all at once, which may result in the loss of information needed for subsequent tokens.
>
>  > Q6. Compare throughput gains at a fixed batch size
>
> Thank you for your suggestion. We have provided some relevant results in our response to Q1.

---

### Official Review · Reviewer_KbMF · 2025-03-12

**Overall Recommendation:** 3

**Summary:**

This paper proposes to offload the KV cache to CPU memory and retrieve KV pairs based on the similarity between the query of a speculative token with quantized KV pairs.

**Claims And Evidence:**

This paper has two claims: 1) Attention is sparse while each token requires different KV pairs. It emphasizes the importance of KV cache offloading. 2) The CPU-GPU communication significantly increases inference latency. These claims are supported by experiments in Fig.2

**Essential References Not Discussed:**

While this paper has cited previous works like [1], the discussion of the difference between the proposed method and previous works is not thorough. Offloading the KV cache to CPU memory is not a new idea. Previous works like [1] also select significant KV pairs based on reduced keys. A discussion of the difference between the proposed method and previous works is lacking.

[1] Tang,J.,Zhao,Y.,Zhu,K.,Xiao,G.,Kasikci,B.,andHan,S. QUEST: Query-aware sparsity for efficient long-context LLM inference. ICML,2024.

**Experimental Designs Or Analyses:**

I checked the experimental results provided in this paper. The experiments are conducted to compare the proposed method with KV cache compression methods with ablation study about the proposed method. However, offloading KV cache to CPU memory is not a new idea, the comparison between the proposed method and the previous methods about KV cache offloading is lacked.

**Methods And Evaluation Criteria:**

The paper proposes to decode an additional speculative token and load KV pairs into the GPU memory based on the speculative token before inference. The method is mainly evaluated on the LongBench dataset.

**Other Comments Or Suggestions:**

I would like to see more experiments and discussion about the difference between the SpeCache and previous KV cache offloading methods. The main contribution of the proposed method is introducing speculative decoding so that the selected KV pairs can be loaded before the decoding step. However, I still have doubts about the improvement of the proposed method over previous methods like QUEST. I look forward to further reply from the authors.

**Other Strengths And Weaknesses:**

Strength:

The speculative token decoding for KV cache preloading seems to be an interesting and novel approach. Generally, the paper is clear and easy to follow.

Weakness:

While offloading the KV cache to CPU memory is not a new idea, there seem to be no experiments comparing the proposed method and previous KV cache offloading methods. While previous work[1] has proposed selecting KV pairs based on reduced keys, a more thorough discussion of the difference and the contribution of this paper is needed.

[1] Tang,J.,Zhao,Y.,Zhu,K.,Xiao,G.,Kasikci,B.,andHan,S. QUEST: Query-aware sparsity for efficient long-context LLM inference. ICML,2024.

**Questions For Authors:**

1) Since each speculative token is decoded based on the previous speculative token, the error may accumulate. As the sequence becomes longer, will the speculative token be less accurate?

2) While it is said to decode two tokens in parallel, does the speculative decoding introduce more latency and GPU memory usage? Are there any experimental results regarding the cost brought by speculative decoding?

**Relation To Broader Scientific Literature:**

This paper focuses on the KV cache offloading technique. There is no relation to broader scientific literature.

**Theoretical Claims:**

As far as I see, there are no theoretical claims in this paper.

---

> ### Author Rebuttal · Authors · 2025-03-31
>
> We thank the reviewer for their thoughtful feedback! We address specific concerns and questions below.
>
> > Q1. Discuss the difference between SpeCache and QUEST and the contribution of SpeCache.
>
> Although both QUEST and SpeCache focus on how to accurately select sparse KV pairs, their emphasis differs due to the distinct scenarios they address.
>
> + QUEST aims to reduce computation and the loading of KV cache **from GPU HBM** by reducing the number of KV pages involved in the attention computation, thereby accelerating the model. In contrast, SpeCache focuses on how to predict and load important KV pairs in parallel when the KV cache is **offloaded to lower-level storage devices (e.g., CPU memory)** for GPU memory saving. Since the CPU-GPU bandwidth (e.g., 16GB/s) is much smaller than the GPU HBM bandwidth (e.g., 1.5TB/s), SpeCache selects a much smaller number of KV pairs to load (64) compared to QUEST (over 1024), and focuses on parallel loading and computation.
>
> + From the experimental results, SpeCache demonstrates a more efficient use of sparse KV pairs. According to the results in QUEST, when the KV budget is less than 1024, model performance significantly declines on longbench (Fig. 7 of the QUEST paper). However, SpeCache performs well even when only 32 KV pairs are loaded (Fig. 5 in our paper). This is because the low-bit KV cache copy we employ can distinguish important KV pairs at a finer granularity, and the low-bit KV cache itself provides some coarse-grained information during decoding.
>
> > Q2. As the sequence becomes longer, will the speculative token be less accurate?
>
> We evaluate 2-bit SpeCache (g=64) and Mistral-7B-Instruct-v0.2 on two long sequence generation tasks, MultiNews and GovReport in Longbench. We track the Speculative Token Top-K KV Cache Hit Rate -- the proportion of the top-k KV cache needed for the next output token that is hit by the top-k KV cache of the speculative token. We separate calculated the hit rate at different decoding step intervals.
>
> |dataset|[1, 10]|[11, 50]|[51, 100]|[101, 200]|[201, ]
> |-|-|-|-|-|-|
> |MultiNews|95.9|92.1|89.1|88.1|87.5|
> |GovReport|97.0|92.5|86.9|87.0|86.8|
>
> We find that as the output grows, the hit rate gradually decreases, but after dropping to around 87%, it stabilizes and maintains a relatively high hit rate.
>
> > Q3. Does the speculative decoding introduce more latency and GPU memory usage?
>
> Since both tokens share the same model weight matrices and KV cache during decoding, speculative decoding neither introduces additional latency from loading pages into the GPU HBM nor increases VRAM usage. Furthermore, as the decoding process is IO-bound and GPU units are not fully utilized, the added computational load does not significantly increase the latency. We verify this through experiments. On Mistral-7B-Instruct-v0.2, we test the latency and VRAM usage of KIVI and KIVI+SpeCache with a batch size of 1 and a context length of 64k.
>
> |Method|Latency (ms/step)|Allocated memory (GB)|
> |-|-|-|
> |Full KV|204.3|31.1|
> |2-bit KIVI (g=64)|94.6|22.8|
> |+ SpeCache|103.6|22.8|
> |1-bit KIVI (g=64)|94.3|22.3|
> |+ SpeCache|101.3|22.3|
>
> We can find that SpeCache only slightly increases the latency of KIVI.

---

> > ### Comment · Reviewer_KbMF · 2025-04-06
> >
> > I appreciate the rebuttal, and I will increase my score

---

> > > ### Author Response · Authors · 2025-04-08
> > >
> > > Thank you very much for your prompt response and recognition of the paper.

---

### Official Review · Reviewer_6JBf · 2025-03-13

**Overall Recommendation:** 4

**Summary:**

This paper presents SPECACHE, a novel method to address the memory bottleneck caused by key-value (KV) caches in large language models when processing long sequences. The authors propose a training-free approach that offloads the complete KV cache to CPU memory while maintaining a low-bit copy in GPU VRAM. The key innovation is a speculative mechanism that predicts which KV pairs will be most relevant for the next token, allowing these to be prefetched from CPU to GPU memory in parallel with ongoing computations. This technique avoids both the information loss associated with compression methods and the latency penalties from offloading approaches. The authors evaluate SPECACHE on LongBench and Needle-in-a-Haystack benchmarks, demonstrating that it can maintain performance comparable to full KV cache while using only 10% of the GPU memory, enabling up to 12x larger batch sizes and 4.6x higher throughput.

**Claims And Evidence:**

The authors' primary claims are well-supported by empirical evidence:
* The claim that KV cache is a memory bottleneck is substantiated with concrete examples in the introduction (page 1, paragraph 2), where they show that for LLaMA 2-7B processing sequences of length 2k with batch size 16, the KV cache size exceeds the model's parameter count.
* The assertion that attention in LLMs is sparse (page 2, paragraph 1) is backed by both references to existing literature and their own analysis in Figure 2 (left), which demonstrates that only 0.5% of keys can cover 90% of a query's attention.
* The claim that SPECACHE maintains model performance while significantly reducing memory usage is well-supported through extensive experiments across multiple models (LLaMA-2, LLaMA-3, and Mistral) on LongBench in Table 1 and Table 2. For instance, with Mistral-7B-Instruct-v0.2, they show a performance gap of only 2% compared to the 16-bit baseline while retaining only 10% of the KV cache size (page 7, paragraph 2).
* The throughput improvements claimed (up to 4.6×) are clearly demonstrated in Table 3 with detailed measurements across different context lengths and batch sizes.

However, I note that the claim about "avoiding inference latency caused by CPU-GPU communication" (page 2, paragraph 2) is slightly overstated. The method mitigates rather than eliminates latency, as shown in Table 4 where parallel prefetching reduces but doesn't eliminate the latency overhead.

**Essential References Not Discussed:**

N/A

**Experimental Designs Or Analyses:**

I checked several aspects of the experimental design and analyses:
* The evaluation on LongBench (Tables 1 and 2) is comprehensive and well-executed, covering multiple models and settings. The authors consistently report average performance across tasks, which gives a clear overall picture.
* The ablation studies (Figures 4 and 5, Table 4 and 5) are well-designed to isolate the impact of specific components. For example, the ablation on 'k' (number of prefetched KV pairs) in Figure 5 shows the trade-off between performance and transfer size.
* The comparison with non-speculative fetching (Table 4) effectively demonstrates the advantage of parallelizing prefetching and computation, showing both performance and latency metrics.
* The throughput measurements in Table 3 appropriately use maximum batch sizes that the GPU memory can handle, providing a realistic assessment of the method's practical benefits.
One analysis that could be improved is the mechanism for selecting the speculative token. On page 4, the authors mention using the output token to compute a speculative token, but don't fully explain how this approximates the next token. More details on this approximation would strengthen the paper.

**Methods And Evaluation Criteria:**

The methods are sound and the evaluation criteria are appropriate for the research question:
* The authors use a comprehensive evaluation approach, testing on the established LongBench benchmark (covering 15 diverse tasks) and the Needle-in-a-Haystack task for specific evaluation of long-context retrieval ability.
* The baseline comparisons are thorough and fair. On page 6-7, the authors compare SPECACHE with several state-of-the-art methods including InfLLM, StreamLLM, H2O, and KIVI with varying compression ratios.
* The experimental setup is clearly described (page 5, section 4.1), specifying implementation details like the number of residual KV pairs and quantization group sizes.
I appreciate the realistic evaluation of throughput (Table 3) across different context lengths, which directly addresses the practical utility of the method. One minor issue is that the authors don't explicitly report statistical significance for their results, though the consistent improvements across multiple models and tasks suggest the findings are robust.

**Other Comments Or Suggestions:**

1. In Figure 3, the illustration is helpful but the distinction between "To be prefetched" and "To be quantized & offloaded" arrows could be clearer.
2. The terminology "SPECACHE" is used inconsistently throughout the paper - sometimes capitalized, sometimes not (e.g., "SpeCache" in Figure 4 caption).
3. On page 8, the reference to "Table 4" in the Ablation section should specify which aspect of Table 4 is being discussed.
4. The paper would benefit from a brief discussion of any overhead in terms of additional CPU computation or memory requirements for managing the offloaded KV cache.
5. Some statements like "without the need for retraining" are repeated multiple times throughout the paper and could be streamlined.

**Other Strengths And Weaknesses:**

Strengths:
* The proposed method is training-free and can be applied to existing pre-trained models without modification, enhancing its practical utility.
* The paper demonstrates impressive results across multiple models (LLaMA-2, LLaMA-3, Mistral) and a range of tasks, showing the broad applicability of the approach.
* The improvement to 1-bit KIVI quantization (page 5, section 3.4) is a valuable contribution in itself, enabling much higher compression ratios than previously possible.
* The method elegantly leverages the memory-IO bound nature of LLM inference to perform speculative decoding with minimal overhead.
* The approach is particularly impactful for long contexts and larger batch sizes, where memory constraints are most significant (Table 3).

Weaknesses:
* On page 4, paragraph 2, the paper states "speculative tokens may be less accurate for output." However, there's no analysis of how often the speculative token differs from the eventual output token, which would help readers understand the approach's limitations.
* The implementation relies on PyTorch's multi-stream mechanism, which the authors acknowledge is not theoretically optimal (page 8, paragraph 1). More details on how a custom implementation could further improve efficiency would strengthen the paper.
* While the authors show SPECACHE works well with KIVI quantization, they don't explore compatibility with other quantization methods like KVQuant or ZipCache, which might provide further improvements.
* The paper lacks discussion of potential failure cases or limitations, such as texts with rapidly changing topics where attention patterns might be less predictable.
* The evaluation focuses exclusively on English text processing. Testing on multilingual settings would provide a more comprehensive assessment of the method's robustness.

**Questions For Authors:**

1. How sensitive is SPECACHE to changes in the distribution of attention patterns? For instance, if the text suddenly changes topic or language, how quickly can the prefetching mechanism adapt?
2. In your experiments, how often did the speculative token differ from the actual next token, and how did this affect the quality of prefetching? This analysis would help quantify the "accuracy" of your speculation mechanism.
3. Your method requires running two forward passes per token generation. Have you explored distilling a smaller model to generate the speculative token, which might reduce computation while maintaining prefetching quality?
4. The CPU-GPU communication is a critical aspect of your approach. How would the performance change with different hardware setups (e.g., PCIe 3.0 vs 4.0, different CPU-GPU bandwidth configurations)?
5. In Figure 2 (middle), you show that decoding latency increases with batch size but remains below fetching latency. Is there a theoretical or empirical upper bound on the batch size where this relationship no longer holds?

**Relation To Broader Scientific Literature:**

N/A

**Theoretical Claims:**

The authors make several theoretical claims that I've verified:
* The analysis of attention sparsity in Figure 2 (left) is sound and aligns with previous findings in the literature. The comparison between query-dependent top-k attention and greedy cache eviction correctly illustrates why dynamic prefetching is necessary.
* The asymptotic analysis of CPU-GPU transfer time (Figure 2, right) correctly shows the linear relationship between transfer size and latency, providing a theoretical foundation for why prefetching only the most important KV pairs is beneficial.
* The assertion that LLM inference is memory-IO bound rather than compute-bound (page 3, end of section 2.2) is supported by both citations and their own measurements, justifying why simultaneous decoding of output and speculative tokens doesn't significantly increase latency.
* The improved 1-bit quantization method (page 5, section 3.4) is theoretically sound, with the modified zero-point and scaling factor ensuring better approximation for uniform distributions.

---

> ### Author Rebuttal · Authors · 2025-03-31
>
> We thank the reviewer for their thoughtful feedback.
>
> > Q1. How sensitive is SpeCache to changes in the distribution of attention patterns?
>
> We add the phrase "How many paragraph are there in the article? Translate the first sentence into German." to the end of a 15k-token text composed of several of Paul Graham's essays, then input it into Mistral-7B-Instruct-v0.2, and observe the output token and speculative token generated via 1-bit SpeCache (black bold text represents the erroneous prediction of speculative tokens):
> + Output token:
> >> There are 13 paragraphs in the article above. The first sentence in German translates to "**Es gibt zwei deutlich unterschiedliche Arten, politisch mittler zu sein: vorsätzlich und zufällig. Absichtslose Mitte stehen dafür, dass die Extreme etwa gleich weit entfernt** sind." (There are two distinct ways to be politically moderate: intentionally and by accident. Accidental moderates are...
> + Speculative token:
> >> are 13 paragraphs in the article above. The first sentence in German translates to "**Ichlieberhnte Wegeliche Arten, politisch mittellangigeben: zufzlich und zufällig." (There are "Intentionenander aus eigentlich Menschen, wenn sie beideehentenfert** sind." (There are two distinct ways to be politically moderate: intentionally and by accident. Accidental moderates are...
>
> We find that when switching languages and topics, speculative tokens diverge from the output tokens within some decoding step. However, as explained in our response to Q2, SpeCache does not rely on the exact match of speculative tokens. It only requires that speculative tokens have a high top-k KV cache hit rate for the next output token.
> > Q2. Accuracy of the speculation mechanism.
>
> We evaluate 2-bit SpeCache (g=64) and Mistral-7B-Instruct-v0.2 on two long sequence generation tasks, MultiNews and GovReport. We track two metrics:
> + Speculative Token Exact Hit Rate: The proportion of speculative tokens that perfectly match the next output token.
> + Speculative Token Top-K KV Cache Hit Rate: The proportion of the top-k KV cache needed for the next output token that is hit by the top-k KV cache of the speculative token.
>
> |dataset|Exact Hit Rate|Top-K KV Cache Hit Rate|
> |-|-|-|
> |MultiNews|57.5|91.3|
> |GovReport|57.6|90.9|
>
> Although the exact hit rate is around 57%, SpeCache does not directly output speculative tokens. Instead, it uses them as a medium to guess the top-k KV cache. The top-k KV cache hit rate is above 90%, which ensures the effectiveness of the speculative tokens.
> > Q3. SpeCache requires running two forward passes per token generation. Have you explored distilling a smaller model?
> + Throughout the inference process, we only add one step of pre-decoding, and in the subsequent decoding phase, SpeCache runs **only one forward pass for each token generation**. This is because SpeCache uses both the Output token (e.g., $T_1$) and the Speculative token (e.g., $T'_2$) as input (i.e., [$T_1$, $T'_2$]) for decoding, which generates two tokens (i.e., $T_1$ generates $T_2$, $T'_2$ generates $T'_3$) in a single forward pass. These tokens are then used as input for the next step, with $T_2$ serving as the model's official output.
> + This is also the reason why the generation of the speculative token is a “free lunch” attached to the forward pass of the output token. Therefore, there is almost no additional overhead, and no need to distill an extra model to generate the speculative token.
>
> > Q4. How would the performance change with different hardware setups.
>
> Due to the varying computational capabilities of different GPUs and the differences in the optimization levels of their operators, it is challenging to conduct a direct ablation study on bandwidth. Therefore, we perform some theoretical analysis to address this. For example, on Mistral-7B-Instruct-v0.2, when the context length is 2K, the maximum batch size for 1-bit SpeCache supported by a 48GB GPU memory is 410. This requires transmitting 3.4GB of KV cache during each decoding step. Theoretically, since SpeCache can fully parallelize data transfer and computation, the decoding latency is 775ms/step. In other words, under optimal conditions, as long as the bandwidth is above 4.4GB/s, the decoding latency of SpeCache will not significantly change.
> > Q5. Theoretical or empirical upper bound on the batch size where "decoding latency < fetching latency" no longer holds?
>
> As the batch size increases, the fetching latency grows linearly, while the decoding latency increases sub-linearly due to improved computation parallelism. This results in the overall decoding latency being lower than the latency for fetching the entire KV cache when the batch size is large. Empirically, we have observed the following behavior: For Mistral-7B-Instruct-v0.2 on an A6000 GPU, with a context length of 2K, when batch size >= 4, fetching latency > decoding latency. Additionally, when the context lengths are 8K or 32K, the boundary conditions shift to 2 and 1, respectively.

---

### Official Review · Reviewer_5qwx · 2025-03-13

**Overall Recommendation:** 2

**Summary:**

This paper proposes storing low-bit KV on the GPU while offloading full-precision KV. Attention is performed between the overall full-precision KV of the top-k keys selected from the low-bit keys.
Additionally, the paper introduces the use of speculative tokens to speculatively prefetch the KV cache needed for the next token. Speculative tokens have been shown to be beneficial for reducing latency.

**Claims And Evidence:**

Most of the claims of this paper are well-supported.
I have one question about table 3: is this throughput run with flash decoding?
I am worried about the slow implementation of torch attention and the overhead burden of two times attention and cpu-gpu io.

**Essential References Not Discussed:**

na.

**Experimental Designs Or Analyses:**

see claims.

**Methods And Evaluation Criteria:**

see claims.

**Other Comments Or Suggestions:**

table 3: throughput -> throughput (tokens/sec)

**Other Strengths And Weaknesses:**

Strengths:

1. This paper is well written with a neat idea.

2. Experiments demonstrate that this method can effectively improve throughput.


Weakness:

1. The improvement w.r.t 2bit KIVI is relatively small, less than 1.5 points. I do not know why the 1-bit improvement is so large. is that due to the 1-bit kivi implementation? I do not think 1 bit is a common setting for kv quant. What is the performance of KVQuant + spec cache?

2. How speculative tokens T2' generated is very confusing. I think it is the main part of this work. Does this spec token work layer by layer? What is pre-generated?

3. I have one question about table 3: Is this throughput run with flash decoding?
I am worried about the slow implementation of torch attention and the overhead burden of two times attention and cpu-gpu io.

Other Suggestions:

1. I suggest that the authors should add a Pareto optimal curve comparison, where the y-axis represents benchmark performance and the x-axis represents the throughput of different methods.
This is because Table 3 does not show a throughput comparison between the authors' method and KIVI, despite KIVI achieving almost the same performance as the authors' results.

2. Add more experiment results on RULER.

3. Add a cache hit rate for speculative tokens.

**Questions For Authors:**

see weakness.

If the author answer these questions during rebuttal, I will improve 2->3.

**Relation To Broader Scientific Literature:**

na.

**Theoretical Claims:**

NA.

---

> ### Author Rebuttal · Authors · 2025-03-31
>
> We thank the reviewer for their thoughtful feedback! We address specific concerns and questions below.
>
>
> > Q1. Why the 1-bit improvement is so large? Performance of KVQuant + SpeCache.
>
> + The improvement of SpeCache on 2-bit KIVI is relatively small because the performance of 2-bit KIVI on longbench is already close to that of 16-bit KV Cache, which can be considered as the performance upper bound after KV Cache compression. As a result, there is limited optimization space left. On the more complex RULER benchmark, SpeCache can provide a significant improvement to 2-bit KIVI (see Q5).
> + In our comparison, all instances of "1-bit KIVI" refer to our improved KIVI implementation described in Section 3.4, since original KIVI can hardly work with 1-bit (see Table 5). The larger improvement of SpeCache on the improved 1-bit KIVI is due to the fact that the improved 1-bit KIVI is still far from the upper bound, leaving more room for optimization.
> + Since the official KVQuant code does not include 1-bit quantization, we compared the performance of SpeCache with 2-bit KVQuant. We conducted experiments on LLaMA-2-7B and measured the PPL on Wikitext-2. The results show that SpeCache also improves the performance of KVQuant.
>
> |Method|PPL|
> |-|-|
> |Full KV|5.12|
> |2-bit KVQuant|7.03|
> |+ SpeCache|5.37|
>
>
> > Q2. Does this speculative token work layer by layer? What is pre-generated?
>
> The pre-decoding phase is a single inference step we introduce between prefilling and decoding. Its purpose is to generate the first speculative token $T’_2$. During pre-decoding phase, $T_1$ is the model input, and the entire model output (i.e., the predicted next token) is used as $T’_2$. In the first decoding step, $T’_2$ is used alongside the $T_1$ as the query, and $T’_2$ will record the top-k KV pairs from each attention layer by layer during decoding, in order to prefetch them for use in the next step.
>
> > Q3. Is this throughput run with flash decoding?
>
> For Full KV Cache, we consistently use Flash-Attention2. For SpeCache and KIVI, during the prefilling phase, we also use Flash-Attention2. However, in the decoding phase, due to the involvement of low-bit operations, we use the low-bit CUDA operators provided by KIVI to compute attention.
>
> > Q4. Throughput comparison between SpeCache and KIVI.
>
> Thank you for your suggestion. Here, we first provide a throughput comparison between SpeCache and KIVI with context length of 32k on Mistral-7B-Instruct-v0.2.
>
> |Method|Throughput (tok/sec)|Performance on Longbench|
> |-|-|-|
> |Full KV|10.3|42.3|
> |2-bit KIVI (g=64)|36.5|39.7|
> |+ SpeCache|34.6|42.0|
> |1-bit KIVI (g=64)|50.8|28.9|
> |+ SpeCache|47.3|40.4|
>
> SpeCache only slightly reduces the throughput of KIVI. Since the computation of speculative tokens is parallelized with the output tokens, the simultaneous decoding of both tokens does not significantly increase the latency.
>
> > Q5. Add more experiment results on RULER.
>
> We evaluate LLaMA-3.1-8B-instruct on RULER with a 32k context length, and the results are as follows:
>
> |Method|KV Size|N-S1|N-S2|N-S3|N-MK1|N-MK2|N-MK3|N-MQ|N-MV|QA-1|QA-2|VT|FWE|Average|
> |-|-|-|-|-|-|-|-|-|-|-|-|-|-|-|
> |16-bit Full KV|1.00x|100.0|100.0|100.0|99.8|99.6|99.6|98.3|98.7|76.8|50.2|98.3|87.1|92.4|
> |2-bit KIVI (g=64)|0.16x|100.0|98.6|93.8|98.8|96.0|66.2|95.7|96.8|72.8|48.2|86.8|86.1|86.7|
> |+ SpeCache|0.16x|100.0|99.8|99.6|100.0|99.0|95.8|98.9|99.0|75.0|49.4|95.7|87.3|91.6|
> |1-bit KIVI (g=64)|0.10x|11.4|12.0|1.8|7.4|2.0|0.0|1.3|0.5|38.0|25.0|2.0|20.1|10.1|
> |+ SpeCache|0.10x|100.0|94.6|94.4|97.2|65.6|34.0|80.6|82.5|59.0|34.7|74.8|74.5|74.3|
>
> We find that 2-bit KIVI experiences significant performance degradation on RULER compared to Full KV, especially on the N-MK3 and VT tasks. However, SpeCache effectively mitigates this issue, keeping the performance gap from Full KV within 1%. While 1-bit KIVI completely fails on all tasks in RULER, SpeCache provides a significant performance boost and even reaches the Full KV cache level on certain tasks, such as N-S1.
>
> > Q6. Add a cache hit rate for speculative tokens.
>
> We evaluate 2-bit SpeCache (g=64) and Mistral-7B-Instruct-v0.2 on two long sequence generation tasks, MultiNews and GovReport in Longbench. We track two metrics:
> + Speculative Token Exact Hit Rate: The proportion of speculative tokens that perfectly match the next output token.
> + Speculative Token Top-K KV Cache Hit Rate: The proportion of the top-k KV cache needed for the next output token that is hit by the top-k KV cache of the speculative token.
>
> |dataset|Exact Hit Rate|Top-K KV Cache Hit Rate|
> |-|-|-|
> |MultiNews|57.5|91.3|
> |GovReport|57.6|90.9|
>
> Although the exact hit rate is only around 57%, SpeCache does not directly output speculative tokens. Instead, it uses them as a medium to guess the top-k KV cache. The top-k KV cache hit rate is above 90%, which ensures the effectiveness of the speculative tokens.

---

> > ### Comment · Reviewer_5qwx · 2025-04-02
> >
> > Considering that FlashDecoding or FlashInfer has effectively become the standard for long-context inference, I will not consider raising the score unless the authors add more experiments about this part, e.g., speccache+fd vs fd. However, I will make sure to keep track of the response throughout the entire rebuttal period.

---

> > > ### Author Response · Authors · 2025-04-03
> > >
> > > Thank you again for your constructive feedback. We provide some clarifications below.
> > >
> > > > Is this throughput run with flash decoding?
> > >
> > > Sorry for not explaining this clearly. In fact, Flash Decoding has already been integrated into FlashAttention2 (see https://github.com/Dao-AILab/flash-attention/issues/1002). When using FlashAttention2 by calling `flash_attn_func` as in our experiments, it heuristically determines whether to execute Flash Decoding.
> > >
> > > To verify this, we replace all `flash_attn_func` calls with explicit `flash_attn_with_kvcache` (i.e., flash decoding) during Mistral-7B-Instruct-v0.2 decoding and test the decoding latency at a sequence length of 64k with batch size = 1:
> > >
> > > |method|latency (ms/step)|
> > > |-|-|
> > > |Full KV ('flash_attn_func')|204.3|
> > > |Full KV ('flash_attn_with_kvcache')|202.8|
> > >
> > > The results show almost no difference between the two, which means that **Flash Decoding was indeed used in our Full KV evaluation**.
> > >
> > > > SpeCache + FD vs FD
> > >
> > > Since our attention design allows the quantized KV cache to participate in computation, it is not supported by official Flash Decoding. However, our customized low-bit kernel, based on KIVI (https://github.com/jy-yuan/KIVI/tree/main/quant), is specifically designed for decoding scenarios, making Flash Decoding unnecessary. To demonstrate this, we also test the inference latency on Mistral-7B-Instruct-v0.2 with a 64k context length and batch size = 1
> > >
> > >
> > > |Method|	Latency (ms/step)|
> > > |-|-|
> > > |Full KV	|204.3	|
> > > |SpeCache (2-bit)|	103.6|
> > > |SpeCache (1-bit)|	101.3|
> > >
> > > It is evident that our low-bit operator is more efficient than Flash Decoding, primarily because the efficient low-bit storage significantly reduces memory access overhead.

---

### Decision · Program_Chairs · 2025-05-01

**Decision:**

Accept (poster)

**Comment:**

This paper introduced an interesting approach to reduce GPU memory during long-context LLM inference by offloading full-precision KV caches to CPU memory. All reviewers agreed that the proposed paper has a novel idea and results are reasonable. There were some disagreement on the on empirical validations. After reading the rebuttal, I believe concerns from reviewer 5qwx was addressed.